

# Merging cranial histology and 3D-computational biomechanics: a review of the feeding ecology of a Late Triassic temnospondyl amphibian

Dorota Konietzko-Meier[1,2], Kamil Gruntmejer[2,3], Jordi Marcé-Nogué[4,5], Adam Bodzioch[2] and Josep Fortuny[5,6]

[1] Steinmann Institute, University of Bonn, Bonn, Germany
[2] Department of Biosystematics, University of Opole, Opole, Poland
[3] European Centre of Palaeontology, University of Opole, Opole, Poland
[4] Centre of Natural History, University of Hamburg, Hamburg, Germany
[5] Virtual Paleontology Department, Institut Català de Paleontologia M. Crusafont, Cerdanyola del Vallès, Spain
[6] Centre de Recherches en Paléobiodiversité et Paléoenvironnements, Muséum national d'Histoire Naturelle, Paris, France

Corresponding author
Dorota Konietzko-Meier,
dorotam@uni.opole.pl

## ABSTRACT

Finite Element Analysis (FEA) is a useful method for understanding form and function. However, modelling of fossil taxa invariably involves assumptions as a result of preservation-induced loss of information in the fossil record. To test the validity of predictions from FEA, given such assumptions, these results could be compared to independent lines of evidence for cranial mechanics. In the present study a new concept of using bone microstructure to predict stress distribution in the skull during feeding is put forward and a correlation between bone microstructure and results of computational biomechanics (FEA) is carried out. The bony framework is a product of biological optimisation; bone structure is created to meet local mechanical conditions. To test how well results from FEA correlate to cranial mechanics predicted from bone structure, the well-known temnospondyl *Metoposaurus krasiejowensis* was used as a model. A crucial issue to Temnospondyli is their feeding mode: did they suction feed or employ direct biting, or both? Metoposaurids have previously been characterised either as active hunters or passive bottom dwellers. In order to test the correlation between results from FEA and bone microstructure, two skulls of *Metoposaurus* were used, one modelled under FE analyses, while for the second one 17 dermal bone microstructure were analysed. Thus, for the first time, results predicting cranial mechanical behaviour using both methods are merged to understand the feeding strategy of *Metoposaurus*. *Metoposaurus* appears to have been an aquatic animal that exhibited a generalist feeding behaviour. This taxon may have used two foraging techniques in hunting; mainly bilateral biting and, to a lesser extent, lateral strikes. However, bone microstructure suggests that lateral biting was more frequent than suggested by Finite Element Analysis (FEA). One of the potential factors that determined its mode of life may have been water levels. During optimum water conditions, metoposaurids may have been more active ambush predators that were capable of lateral strikes of the head. The dry season required a less active mode of life when bilateral biting is particularly efficient. This, combined with their characteristically anteriorly positioned orbits, was optimal for

ambush strategy. This ability to use alternative modes of food acquisition, independent of environmental conditions, might hold the key in explaining the very common occurrence of metoposaurids during the Late Triassic.

# INTRODUCTION

Temnospondyli is one of the most diverse groups of early tetrapods, which flourished worldwide during the Carboniferous, Permian and Triassic periods and survived the Triassic-Jurassic extinction as relics in eastern Asia and Australia until the Early Cretaceous (*Holmes & Carroll, 1977*; *Milner, 1990*; *Warren, Rich & Vickers-Rich, 1997*; *Schoch, 2013*). The most characteristic and best-known part of the temnospondyl skeleton is the skull. This is a flat structure with few fenestrae on the skull roof (nares, orbits and, in some capitosaurs, the closed otic notch); the palatal side has more extensive openings: large subtemporal windows, interpterygoid vacuities and choanae. Despite extensive studies and numerous fossil records, lots of issues of temnospondyl biology and mode of life still remain unclear. One crucial issue concerns their mode of feeding. Temnospondyls were carnivorous, but whether they mainly used suction feeding and/or direct biting is still unclear (*Milner & Sequeira, 1998*; *Warren, 2000*; *Steyer et al., 2006*; *Witzmann, 2006*; *Damiani et al., 2009*; *Maganuco et al., 2009*; *Fortuny et al., 2011*).

A useful method for understanding form-function relationships is FEA. Over recent years, FEA has been intensively used to study the biomechanical behaviour of a wide array of vertebrates, providing new insights into the exploration of function and morphological evolution, particularly adaptation and biological structural constraints (*Rayfield, 2007*). FEA is used to obtain stress distribution patterns that can be interpreted in order to understand either mechanical behaviour or evolutionary adaptation (*Witzel et al., 2011*). A number of studies have used computational biomechanical analyses to address the question of feeding strategies among Temnospondyli (*Stayton & Ruta, 2006*; *Fortuny et al., 2011*; *Fortuny et al., 2012*; *Fortuny et al., 2016*; *Marcé-Nogué et al., 2015*; *Lautenschlager, Witzmann & Werneburg, 2016*; *Fortuny, Marcè-Noguè & Konietzko-Meier, 2017*). One such method, finite element analysis (FEA), documents deformation and distribution of strains and stresses in the skulls that are related to different ecomorphologies (*Fortuny et al., 2011*; *Fortuny et al., 2012*; *Fortuny et al., 2016*; *Marcé-Nogué et al., 2015*; *Lautenschlager, Witzmann & Werneburg, 2016*; *Fortuny, Marcè-Noguè & Konietzko-Meier, 2017*).

However, computational modelling requires numerous methodological assumptions and simplifications which can lead to inaccuracies or misinterpretations. This is especially true for fossil taxa because of the general inability to compare the predicted scenario with results obtained from living animals. A common oversimplification is to limit the number of biomechanical scenarios tested (for a discussion, see *Fortuny et al., 2015*). For complex functions, such as feeding, it is important to have scenarios additional to those performed

in FEA analyses. Another problem involves oversimplifications of boundary conditions and mechanical bone properties, whether or not bones are modelled with elastic linear, homogeneous material properties, which are calculated by using known values for the entire structure (*Anderson et al., 2012*; *Bright, 2014*). For fossil taxa without modern analogues these variables have to be assumed on the basis of taxa with a similar Bauplan, even if these are only distantly related (*Anderson et al., 2012*) as for Temnospondyli (*Sanchez et al., 2010*; *Fortuny et al., 2016*) which belong to amphibians and for which commonly the genus *Crocodylus* is used as a proxy (see a discussion, see *Fortuny et al., 2016*). Moreover, histological studies of metoposaurid skulls have shown that the histological framework of skull bones is very variable (*Gruntmejer, Konietzko-Meier & Bodzioch, 2016*), which has never been taken into account during FEA analysis of fossil taxa.

This implies making assumptions regarding bone properties and input conditions (see *Bright, 2014*; *Fortuny et al., 2015* and references therein). The influence of these oversimplifications of model construction has not yet been fully examined and is therefore poorly understood. Thus, validation and testing the reliability of results using sensitivity analyses is necessary (i.e., *Ross et al., 2005*; *Kupczik et al., 2007*; *Wang et al., 2010*; *Wang et al., 2012*; *Bright & Rayfield, 2011*; *Cox et al., 2011*; *Wood et al., 2011*; *Bright, 2012*; *Fitton et al., 2012*; *Walmsley et al., 2013*; *McCurry, Evans & McHenry, 2015*). However, sensitivity analyses of a model can only be carried out for extant taxa by comparing the model against data collected *in vivo/in vitro*. This is impossible for fossil taxa, especially for those without any homologous taxa among closely related extant relatives, such as Temnospondyli.

For such cases one approach to ensure reliable results may be bone histology. The bony framework is a product of biological optimisation and bone structure is created to meet local mechanical conditions. Bone microstructure can be used to estimate local stress. Biomechanical properties of bone histology have been extensively analysed for a considerable time (i.e., *Martin, 1991*; *Currey, 2003*; *Currey, 2006*; *Currey, 2012*; *Currey, Pitchford & Baxter, 2007*; *Zioupos, Hansen & Currey, 2008*; *Mishra, 2009*; and references therein). Bone microstructure is related directly to loads and can be modified during the animal's life time (short-term adaptation) and/or on the long term, as an evolutionary adaptation. The mechanical properties of bone are the result of a compromise between the need for a certain stiffness (i.e., to reduce deformation in the bone and achieve more efficient kinematics) and the need for enough ductility to absorb impacts (i.e., to reduce the risk of fracture and minimise skeletal weight), while maintaining adequate biological safety factors (*Biewener, 1993*). The strength of a structure is the product of organisational and compositional features (*Currey, 2012*). With regard to bone microstructure, the most important organisational feature is porosity, because bone loses strength and stiffness with increased porosity. This is explained by the fact that soft tissues have essentially no strength or stiffness with respect to non-hydrostatic stresses (*Martin, 1991*). However, high cortical thickness can compensate for low resistance of bone tissue (*Carrier & Leon, 1990*; *Margerie et al., 2004*). Compact bone is associated with high strength in tension, but accompanied by a lack of strength in compression, which is higher for trabecular bone (*Martin, 1991*; *Currey, 2003*; *Rhee et al., 2009*; *Achrai & Wagner, 2013*; and references therein).

A significant portion of the skeleton of early amphibians consisted of dermal bone (skull, mandible, clavicle and interclavicle), either intramembranous or metaplastic in origin. Dermal bone, as a specific combination of trabecular and cortical bone, forms a "sandwich-type" or plywood structure which is well known in engineering for its optimum structural properties (*Currey, 2006*). In large flat bones which are bent along their shortest dimension, the cancellous bone forms the middle of a sandwich, with the compact shell bearing the major loads and the cancellous bone keeping the walls of the shell apart and dealing with any shearing loads that may arise. Moreover, dermal bone texture provides substantial increase in strength and stiffness that is accompanied by a relatively small increase in mass (*Witzmann, 2009*; *Rinehart & Lucas, 2013*). Calculations have demonstrated that there is a property/mass advantage, albeit a modest one, in having cancellous bone in the middle, rather than having a solid, but overall thinner bone (*Currey, 2006*; *Currey, 2012*). A mechanical advantage of metaplastic bone is the firm connection between bone and overlying soft tissue, since the collagen fibres of the attached soft tissue are confluent with the collagen fibres within the metaplastic bone (*Haines & Mohuiddin, 1968*).

To test whether or not FEA models and histological results provided similar predictions of cranial mechanical behaviour under feeding loads, the present paper compares results from both methods for a well-known taxon. The early tetrapod *Metoposaurus krasiejowensis* (*Sulej, 2002*) (Metoposauridae, Temnospondyli) from the Upper Triassic of southwest Poland provides an interesting case study in view of the great number of excellently preserved specimens recovered as well as the extensive data set for this taxon (*Sulej, 2002*; *Sulej, 2007*; *Barycka, 2007*; *Konietzko-Meier & Klein, 2013*; *Konietzko-Meier & Sander, 2013*; *Gruntmejer, Konietzko-Meier & Bodzioch, 2016*; *Fortuny, Marcè-Noguè & Konietzko-Meier, 2017*; *Teschner, Sander & Konietzko-Meier, 2017*).

In the present study a new concept of using bone microstructure to predict stress distribution in the skull during feeding is outlined and a correlation between predictions from bone microstructure and computational biomechanics (FEA) results is carried out as well. Merging results from these two methods will first test predictions generated by FEA and help to evaluate the influence of methodological assumptions and simplifications on the final results and, secondly, yield new insights into the feeding ecology of Temnospondyli.

## MATERIAL AND METHODS

### Material

Two skulls of *Metoposaurus krasiejowensis*, housed in the collections of Opole University (UOPB), were analysed. One of these (UOPB 00124; 290 mm in length) was CT scanned for 3D-Finite Element Analysis, while the second (UOPB 01029; 400 mm in length) was studied histologically (Fig. 1). Both specimens were collected at Krasiejów, the Upper Triassic locality in southern Poland (of Norian age, according to recent stratigraphical studies: *Racki & Szulc, 2015*; *Szulc, Racki & Jewuła, 2015*; *Szulc et al., 2015*).

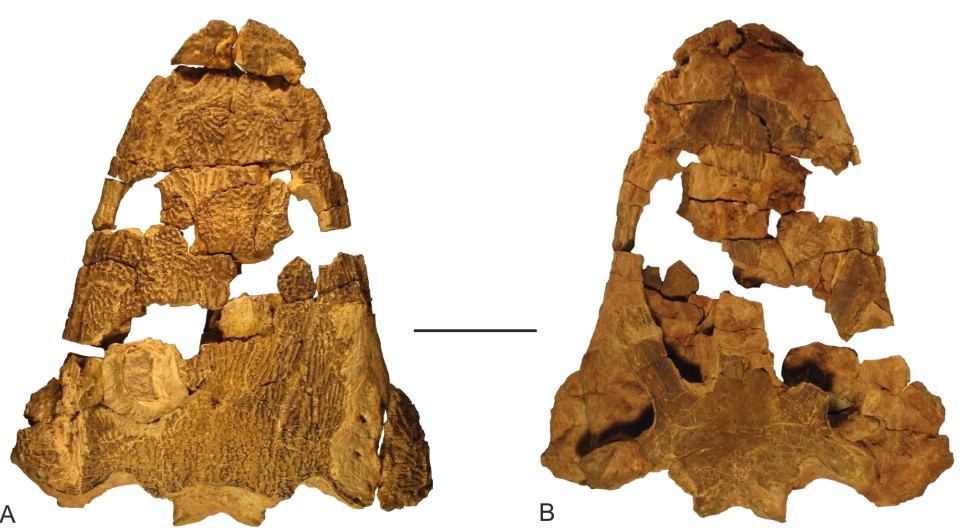

**Figure 1** **Skull of *Metoposaurus krasiejowensis* from the Upper Triassic of southwest Poland (UOPB 01029) used in the histological study, in dorsal (A) and palatal (B) views.** Scale bar equals 10 cm.

## Methods
### *Finite elements analysis*

Skull UOPB 124 of *Metoposaurus krasiejowensis* was CT scanned at the Hospital Mutua de Terrassa (Catalonia, Spain), using a medical CT scanner Siemens Sensation 16. The specimen was scanned at 140 kV and 150 mA, obtaining a 0.576 mm pixel size and an output of $512 \times 512$ pixels per slice with an interslice space of 0.3 mm. The specimen corresponds to an entire skull, completely matrix filled. After segmentation, a 3D model only of the skull was generated. During this last step, the surface irregularities were repaired using refinement and smoothing tools from Rhinoceros 5.0 software and imported into ANSYS 16.2 software to perform FEA (see *Fortuny, Marcè-Noguè & Konietzko-Meier, 2017* for further details).

A Structural Static Analysis to evaluate the biomechanical behaviour during biting was performed using the Finite Element Package ANSYS 16.2 in a Dell Precision[TM] Workstation T7600 with 32 GB ($4 \times 8$ GB) and 1600 MHz. Elastic, linear and homogeneous material properties were assumed for the bone using the following values: *E* (Young's modulus): 6.65 GPa and m (Poisson's ratio) 0.35 (*Currey, 1987*), from frontal bone of *Crocodylus*. The skull of *Metoposaurus krasiejowensis* was meshed with an adaptive mesh of hexahedral elements (*Marcé-Nogué et al., 2015*). The mesh of the model was around 2.2 million elements and 3.1 million nodes. A gape angle of 10° was used, although the model was tested also under 20° gape angle, obtaining a very similar distribution pattern (see *Fortuny, Marcè-Noguè & Konietzko-Meier, 2017* for a comparison).

Four loading cases were analysed considering bilateral, unilateral, lateral prehension/bite and skull-raising system (Fig. 2). The bilateral case simulates a bilateral bite on both left and right sides of the skull, whereas the unilateral case simulates the same bite only on the right side. The lateral case simulates a lateral loading direction to generate a within-plane

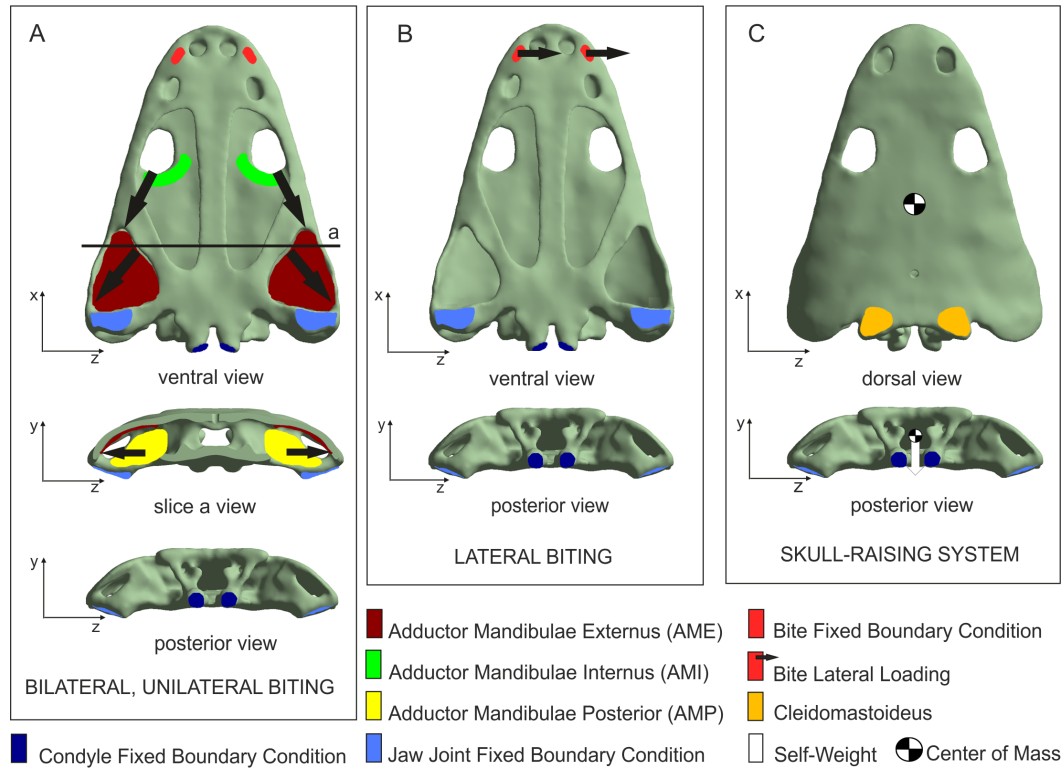

**Figure 2  Loading and boundary conditions used to simulate (A) bilateral and unilateral biting, (B) lateral biting, and (C) skull-raising system.** Coloured areas indicate muscle insertion areas (AME, AMI, AMP and Cleidomastoideus, respectively). Arrows indicate the direction of the muscle force applied. Adapted from *Fortuny, Marcè-Noguè & Konietzko-Meier (2017)*. Image credit: *Journal of Anatomy*/Wiley.

lateral bend to the snout and simulate movement of the head through the water, assuming that *Metoposaurus* could have hunted prey by using a rapid lateral sweep of the head during active swimming. The skull-raising case considered the motion of the skull (relative to the lower jaw) when the mouth opens. All cases are illustrated in Fig. 2 and explained in more detail by *Fortuny, Marcè-Noguè & Konietzko-Meier (2017)*.

Displacements at the jaw joint were restricted in the $y$-direction, simulating the contact with the jaw and near the double-headed occipital condyle in the $x$-direction, simulating the presence of the vertebral column.

For the bilateral and the unilateral cases, forces exerted by the Adductor Mandibulae Internus (AMI), the Adductor Mandibulae Externus (AME) and the Adductor Mandibulae Posterior (AMP) were considered in the model according to soft-tissue reconstruction based on several authors (e.g., *Carroll & Holmes, 1980*; *Sulej, 2007*; *Steyer, Boulay & Lorrain, 2010*; *Witzmann & Schoch, 2013*; *Marcé-Nogué et al., 2015*; *Fortuny et al., 2016*; *Fortuny, Marcè-Noguè & Konietzko-Meier, 2017*). The muscular insertion areas of AMI, AME and AMP were defined in the model in order to apply the forces of muscular contraction during different prehension/bites. The direction of these forces was defined by the line that joins the centroid of the insertion area in the skull with its correspondence in the insertion area

of the lower jaw. Following *Alexander (1992)*, a value of 0.3 MPa was assumed as muscular contraction pressure. This resulted in applied muscle forces of 229.7 N for the AMI, 1963.9 N for the AME and 685.2 N for the AMP. To simulate biting, a fixed boundary condition in the three dimensions ($x$, $y$ and $z$) was applied in the bite location to simulate the moment that skull and mandibles contact the prey.

For the lateral case, an arbitrary force of 100 N was applied in the $z$-direction at the position of the fangs in the palate. Finally, for the skull-raising case, a muscular force was applied on the cleidomastoideus muscle creating null displacement of the tip of the snout when the overall weight of the skull is applied.

### Thin sections

A histological study of cranial bones of *Metoposaurus krasiejowensis* (UOPB 01029) has indicated a relatively stable collagen fibre pattern with parallel-fibred bone constructing the grooves and inner cortex and lamellar bone present in the troughs/grooves of the skull (*Gruntmejer, Konietzko-Meier & Bodzioch, 2016*). In contrast, microstructural characters (thickness and compactness) change very clearly (*Gruntmejer, Konietzko-Meier & Bodzioch, 2016*), and thus may be used as a proxy to estimate the mechanical loading. However, detailed studies were not performed to analyse the relations between thicknesses, compactness and estimated biomechanical loading. This is why these two features, based on the same thin section collection as published by *Gruntmejer, Konietzko-Meier & Bodzioch (2016)* are analysed here.

UOPB 01029 was sectioned in 16 places, inclusive of 17 flat dermal bones (Fig. 3; Table 1), according to standard petrographical procedures (*Chinsamy & Raath, 1992*). Non-dermal bones such as the exoccipital and quadratojugal were not analysed, because of their endochondral origin and different shape. Subsequently, thin sections were studied under a LEICA DMLP light microscope in normal and polarised light.

In the thin sections, the average thickness of the entire bone was estimated, expressed as an arithmetical average from three measurements of the thickness of the entire bone. The average thickness of the bone was measured three times over the distance between the ventral side of the bone and the bottom of troughs/grooves, and three times as the distance to the top of ridges. The mathematical average was calculated from these measurements.

To estimate bone porosity the thin sections were scanned using an Epson Scanner and transformed into black and white images. The analysis of compactness was done using software Bone Profiler (*Girondot & Laurin, 2003*).

In the prefrontal, squamosal 2 and parasphenoid, on account of the strongly variable bone thickness, the significantly thinner parts of these bones (labelled in Figures and in the text as -b, as opposed to the thicker part named -a), were calculated separately.

## RESULTS

### Cranial biomechanics based on finite element analysis

Values of equivalent Von Mises stresses and their distribution were recorded in order to compare their behaviour under the effect of loads and constraints in the bilateral,

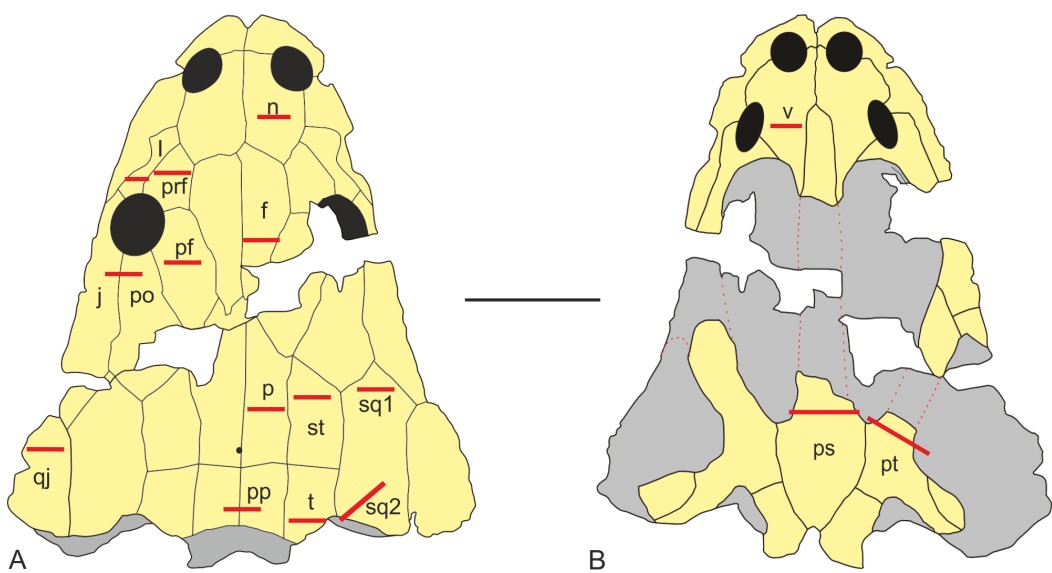

**Figure 3** **Sectioning planes of dermal bones of skull of *Metoposaurus krasiejowensis* (UOPB 01029) in dorsal (A) and palatal (B) views.** Scale bar equals 10 cm. Abbreviations: f, frontal; j, jugal; l, lacrimal; n, nasal; p, parietal; pf, postfrontal; po, postorbital; pp, postparietal; prf, prefrontal; ps, parasphenoid; pt, pterygoid; qj, quadratojugal; sq1, squamosal-1; sq2, squamosal-2; st, supratemporal; t, tabular; v, vomer.

unilateral, lateral biting and skull-raising for simulating feeding behaviour (see also *Fortuny, Marcè-Noguè & Konietzko-Meier, 2017*).

Under bilateral biting, the model showed moderate to low-level stresses on most parts of the skull, with just a few peak stress levels in the posterior part (Fig. 4A). Small spots of stress were present on the dorsal portion of the supratemporal and posterior part of the squamosal, but mainly in ventral portions of the jugals and supratemporal and the posterior ramus of the pterygoid. Of particular interest is the absence of stress around the premaxilla, the posterior part of the maxilla and lacrimal and the naso-frontal region. Stress slightly increased around the orbits. On the palate, a few peak levels of stress were present near the choanae, with no levels of stress on the premaxilla, nor on most of the cultriform process and the parasphenoid.

Simulating unilateral biting reveals a different stress pattern for left and right parts of the skull with significantly higher levels of stress on the right side (Fig. 4B). The preorbital region presents moderate to high levels of stress in the region of anterior part of the right maxilla and increasing of loading in the interorbital region. An extremely high level of stress is reconstructed in the posterior part of the parietals and the postparietals as well as the otic notch (posterior part of the squamosal) up to the quadratojugal. The predicted stress level for the palate is even higher and refers to the quadratojugal-quadrate region and most of the pterygoid and is particularly high in the parasphenoid, and most of the cultriform process, as well as the exoccipitals (Fig. 4B).

The general pattern during lateral loading revealed low or absent levels of stress on the skull roof; on the quadratojugal stress was low (Fig. 4C). It is particularly significant that

**Table 1** The thickness and compactness of the dermal skull bones of *Metoposaurus krasiejowensis* from Late Triassic of Poland.

| Bone | Average thickness (μm)[a] | Compactness (%)[b] |
|---|---|---|
| Nasal | 4,758 | 77.4 |
| Prefrontal | 4,256 | 78.5 |
| Lacrimal | 5,940 | 79.2 |
| Frontal | 5,549 | 76.3 |
| Postfrontal | 3,960 | 94.5 |
| Jugal | 6,940 | 87.8 |
| Postorbital | 6,940 | 82.6 |
| Parietal | 3,840 | 92.2 |
| Supratemporal-a | 4,800 | 92.5 |
| Supratemporal-b | 2,050 | 98.6 |
| Squamosal-1 | 3,915 | 89.1 |
| Squamosal-2-a | 4,000 | 94.0 |
| Squamosal-2-b | 1,600 | 98.7 |
| Quadratojugal | 5,610 | 78.6 |
| Postparietal | 8,670 | 80.7 |
| Tabular | 10,000 | 82.5 |
| Vomer | 2,925 | 54.7 |
| Parasphenoid-a | 4,050 | 77.7 |
| Parasphenoid-b | 2,100 | 91.0 |
| Pterygoid | 5,460 | 73.1 |

Notes.

[a]The average thickness of entire bone was estimated in thin sections, expressed as an arithmetical average from three measurements of the thickness of a bone taken on the bottom of valleys and the top of ridges.

[b]Compactness was estimated using the software Bone Profiler (*Girondot & Laurin, 2003*).

the antorbital region had extremely low levels of stress. However, on the palate, the general stress levels increase: low or moderate levels were seen on the vomer and premaxilla, while the cultriform process presented low levels in its anterior part, moderate ones in its posterior part and an absence from the central part. As far as the posterior part of the skull is concerned, high stress levels were present on the posterior branch of the pterygoid and quadratojugals, while the anterior part of the pterygoid had low or very low stress levels under lateral loading, while its central area (adjacent to the parasphenoid) revealed moderate levels of stress. The major part of the parasphenoid had moderate and high levels of stress, increasing on the posterior part of the parasphenoid and in the exoccipitals.

The simulation of the skull raising system during jaw opening showed that the stress values are very low along nearly the entire skull (Fig. 4D). The stress values increase only significantly in the interorbital region, the regions around the otic notches, and the central part of the cultriform process as well as posterior rami of the pterygoids (Fig. 4D). Lower, but still measurable, stress is indicated in the anterior rami of the pterygoid, ectopterygoid and in the vomer (Fig. 4D).

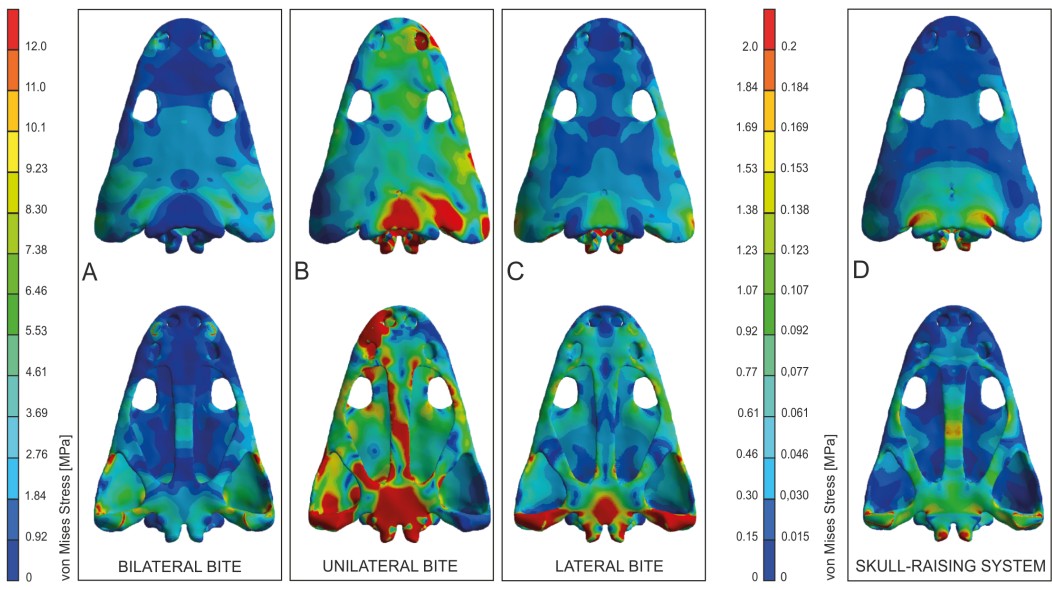

**Figure 4** **Von Mises stress results (in MPa) of bilateral (A), unilateral biting (B), lateral (C) and of skull raising system (D) in *Metoposaurus krasiejowensis* (UOPB 00124) using a gape angle of 10°.** Adapted from *Fortuny, Marcè-Noguè & Konietzko-Meier (2017)*. Image credit: *Journal of Anatomy*/Wiley.

## Biomechanical loading approach from thin section analysis
### Bone microstructure

The average thickness of dermal bones varies from 2 to 10 mm (Table 1, Figs. 5 and 6). In the posterior part of the skull, bones are the thickest (postparietal and tabular), up to 10 mm and a compactness varying between 80 and 82%. The microstructural characters suggest a very high biomechanical loading on this part of the skull (Table 1; Fig. 5). The postorbital and jugal represent a similar average thickness (close to 7 mm), but the compactness varies from 83 to 88%, respectively (Table 1; Fig. 5) and thus the postorbital may have been less loaded. However, bone microstructure predicts a lower loading, compared to the posterior part of skull, but still both bones have high stress levels. Further decrease of the strength is observed for the lacrimal, quadratojugal, frontal and pterygoid bones, in which the thickness oscillates around 6 mm and the compactness varies from 73 to 80% (Table 1; Figs. 5 and 6). The postfrontal, squamosal-1, squamosal-2-a, parietal and supratemporal-a present thicknesses of around 4 mm, with the compactness changing from 88 to 95% (Table 1; Figs. 5 and 6). The postfrontal, squamosal-1, squamosal-2-a, parietal and supratemporal-a are thinner as lacrimal, quadratojugal, frontal and pterygoid, but their compactness is considerably higher and for all these bones a moderate stress level could be predicted (Table 1; Figs. 5 and 6). The nasal, prefrontal and parasphenoid-a with a relatively limited thickness and compactness received low biomechanical loads (Table 1; Figs. 5 and 6). The vomer is extremely porous, with a compactness of only about 55%, accompanied by limited thickness and possibly free of stress (Table 1, Figs. 5 and 6). Markedly thinner and with a low load appear also medial parts of the supratemporal and parasphenoid (in Figs. 5 and 6 marked as -b) reaching about 2 mm of thickness, contrary

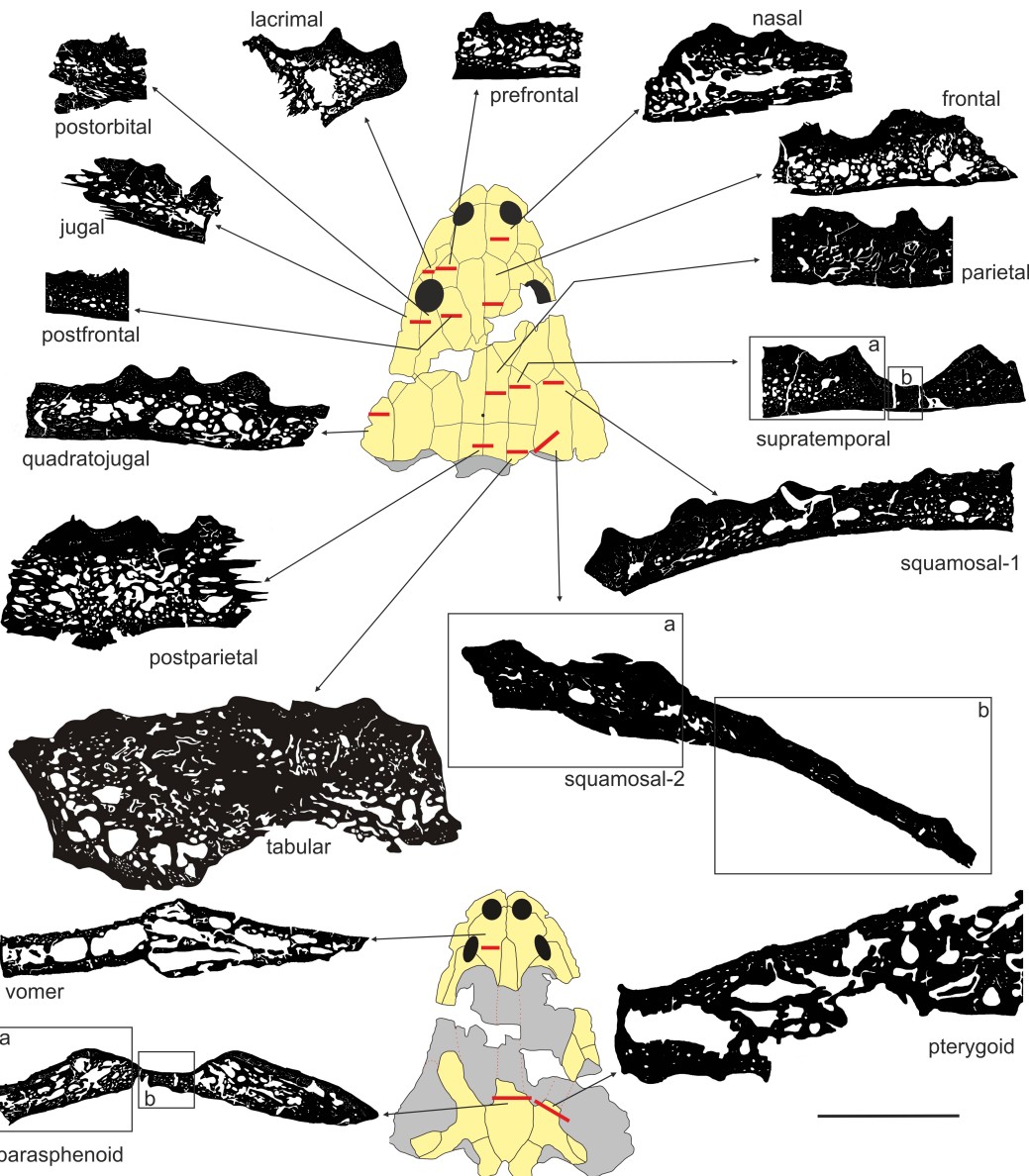

**Figure 5** **General microstructure of skull bones in *Metoposaurus krasiejowensis* (UOPB 01029).** All bones are shown in the same scale, whereas the skull miniatures are included only to show the position of bones and are not in scale. Scale bar equals 10 mm.

to the remaining parts (supratemporal-a and parasphenoid-a) with an average thickness of about 4 mm (Table 1, Figs. 5 and 6). In squamosal-2 the change in bone thickness is more gradual (Fig. 5). The supratemporal-b, parasphenoid-b and squamosal-2b are the thinnest among all bones, albeit are extremely compact, over 90% (Table 1; Figs. 5 and 6).

To summarise, there is a linear relationship (Fig. 6) between compactness and average bone thickness, with the only exception of vomer, being an outlier. Bones with high

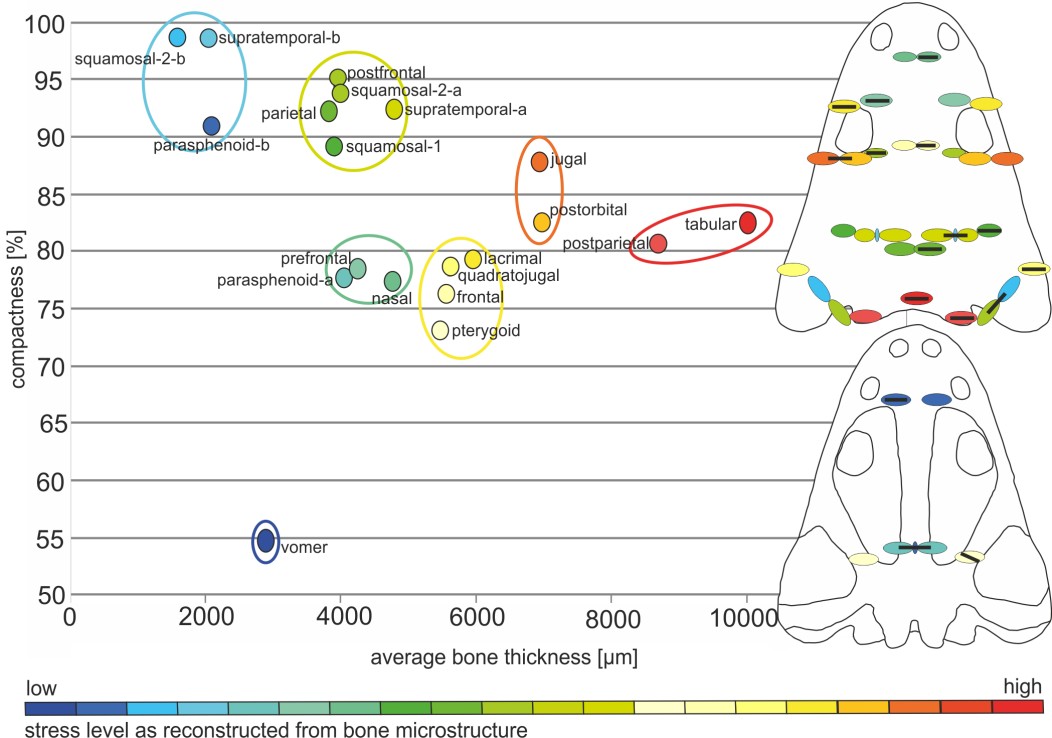

**Figure 6** **Estimated biomechanical loading as reconstructed on the basis of microstructural characters of the skull of *Metoposaurus krasiejowensis* (UOPB 01029).** Note that the estimated values (average bone thickness *vs* compactness) are relative and show merely if loading on any given region was higher or lower (on the scale bar from red to blue, respectively). It is not possible to calculate the objective amount of stress in this case. Black bars inside colour ellipses indicate the sectioning places; the ellipses without bars are symmetrical to sectioned areas.

compactness have low values of thickness and bones with low values of compactness have higher values of thickness.

### Cranial biomechanical loading approach from bone microstructure

The microstructural bone characters suggest a high biomechanical resistance of the posterior part of the skull; this is moderate in the preorbital and along the lateral edges of the skull roof, with a tendency to decrease in the otic notch region and postorbital area (Fig. 6). A slight increase of biomechanical loading is present next to the posterior margin of the orbits (Fig. 6). The squamosal-2-b which is the anterior part of this bone and a narrow valley that represents a lateral canal in the supratemporal (supratemporal-b) are considerably weaker than the remaining part of these bones (Fig. 6). Squamosal-2 shows a gradual increase in mechanical resistance in a posterior direction, predicting slightly higher loading on the lateroposterior part (squamosal-2-a) of the skull (Fig. 6). The postfrontal, squamosal-1, posterior part of squamosal-2 (squamosal-2-a), parietal and supratemporal-a are thinner than the lacrimal, quadratojugal as well as frontal, but significantly more compact and thus represent only a slight decrease of biomechanical strength. With the same values in respect of thickness as in the prefrontal, nasal and parasphenoid-a, the

postfrontal, squamosal-1, squamosal-2-a, parietal and supratemporal-a are more compact and thus able to resist higher stress (Fig. 6).

The microstructure of bones from the palatal side suggests moderate or low stress levels. The vomer and the medial part of the parasphenoid (Fig. 6—parasphenoid-b) appear to be extremely weak, whereas the anterior branch of the pterygoid shows greater biomechanical resistance.

## DISCUSSION

### New insights into skull biomechanics—merging methods

As a proxy to test predictions from FEA simulation, bone microstructure was used. In this context, very low levels of Von Mises stress appear in the thin and porous bones (vomer) and these levels increase when the compactness of the bone is higher and the thickness lower (parasphenoid-b, squamosal-2-b and supratemporal-b). As a general trend, high values of Von Mises stress appear in thicker bones, such as tabular, jugal or postorbital (Fig. 7).

The FEA results demonstrated that metoposaurids preferred rapid bilateral biting along with lateral strikes of the head, even if then latter behaviour was not preferred (*Fortuny, Marcè-Noguè & Konietzko-Meier, 2017*). Based on FEA results, unilateral biting was avoided because the skull would experience a comparatively very high stress level, probably due to the absence of a secondary palate (*Fortuny et al., 2016*; *Fortuny, Marcè-Noguè & Konietzko-Meier, 2017*). The histological framework confirms a very close stress distribution pattern obtained during FE analyses (Fig. 7), including the fact that it was very far from optimal and not efficient for metoposaurs to perform unilateral biting in any scenario (Fig. 7A). The only case for which the histological model and unilateral FEA loading show the same tendency is a high loading present in the posterior part of skull (Fig. 7A) whereas under bilateral and lateral FEA loadings, this skull region receives a low or moderate level of stress (Figs. 7B and 7C). However, in this case the similar signal in unilateral and histological models is only a methodological artefact. The microstructural analysis reveals for the tabular and the postparietal that these elements are biomechanically adapted to receive high amounts of stress (Figs. 5 and 6). Moreover, these bones are strongly metaplastic, which suggests a tight connection to muscles or ligaments (*Gruntmejer, Konietzko-Meier & Bodzioch, 2016*). The reason for the presence of extremely resistant bones in this region is not directly connected with biting itself. A slight increase of stress level visible in the FEA model of skull raising (Fig. 7D) suggests that other variables, which are related directly to the mouth opening, affected the tabular and parotic process, otic notch and mainly the cleidomastoideus muscles and could explain the strength and biomechanical capabilities found in the histological analysis.

The microstructural characters confirm that the nasal and prefrontal are relatively weak bones; thus, the biomechanical loading in these regions was relatively low. In both simulations (bilateral and lateral) the same tendency is observed, with a slight increase of stress level in the prefrontal (Figs. 7B and 7C), which is also thinner (Fig. 5). In the frontal region, under a bilateral case, the estimated stress level exceeded that in the prefrontal and
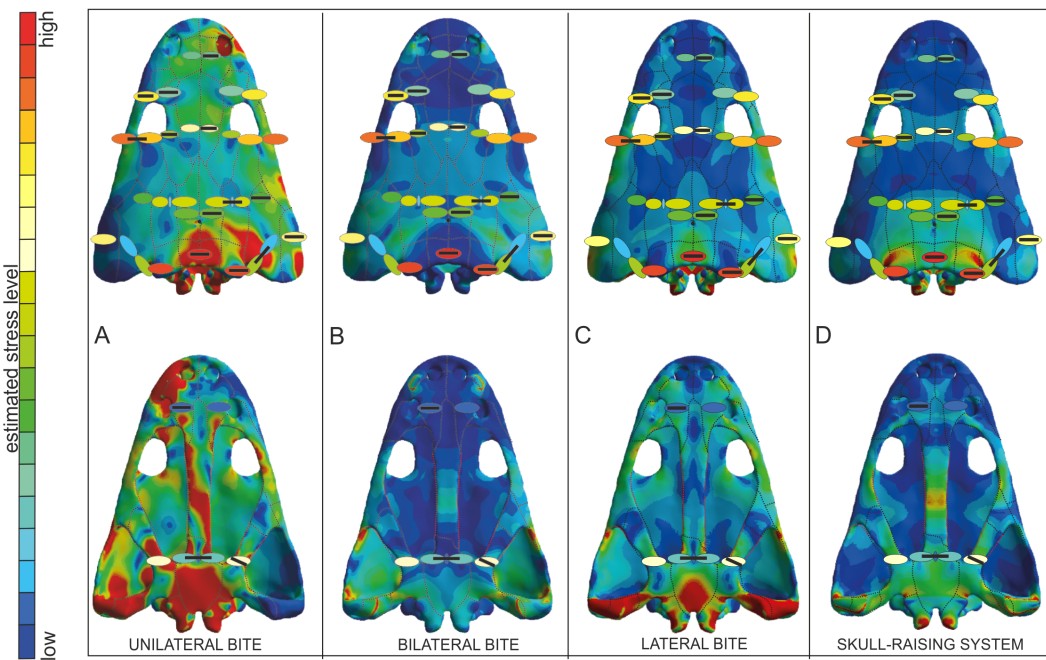

**Figure 7** **Von Mises stress results merged with the model of biomechanical loading created on the basis of microstructural characters.** Von Mises stress results represent unilateral (A), bilateral (B), lateral (C) biting and of skull-raising system (D) in *Metoposaurus krasiejowensis* (UOPB 00124) using a gape angle of 10°. Model of biomechanical loading is created on the basis of microstructural characters for skull UOPB 01029. Skulls in the background show FEA results; the outcome of histological reconstructions is illustrated as oval forms, of different colours. Note that similar colours were used in FEA analysis and histological estimates in order to illustrate how the general stress distribution in FEA and histological analyses correlate; however, identical colours do not signify identical stress values. In FEA, the colours refer to objective values (compare with Fig. 4), while in histological models estimated values are relative and show merely if the loading on any given region is higher or lower within a single skull (on the scale bar from red to blue, which means from high to low loading, respectively). Black bars inside colour ellipses indicate the sectioning places; the ellipses without bars are symmetrical to section areas. The black dotted lines shows the sutures, the red lines indicate the reconstructed borders.

nasal (Fig. 7B). Histologically, the frontal is a relatively massive bone in comparison to the nasal and prefrontal. This indicates that loading on the frontal region was probably high. The postfrontal, squamosal-2-a, parietal and squamosal-1 are thinner than the frontal, but more compact and thus the loading could be the same or only slightly lower as on the frontal (Figs. 5 and 6). The same tendency is observed in bilateral and lateral FEA cases (Figs. 7B and 7C). Squamosal 2 is of special interest as the FE models suggests a change of stress value from the posterior to the anterior part of this element, receiving high to low levels, respectively (Fig. 7). Interestingly, the histological results also reveal this change in the thickness of the cortex as well as in the porosity across the bone (Fig. 6). Squamosal-2-b in both FEA cases (bilateral and lateral) show barely any stress anteriorly (Figs. 7B and 7C); moreover, histologically, squamosal-2-b is a very thin bone, indicating that the mechanical loading of this part of skull was very low, only slightly increased posteriorly (Fig. 5). As an entire bone, the supratemporal is adapted to moderate loading (Fig. 6), which is visible also

in bilateral and lateral FEA cases (Fig. 7). The histological framework of the supratemporal suggests that the most sensitive part of this bone, which represents a drastically reduced strength, is the lateral line canal (Fig. 6). The increase in bone compactness visible in the canal might partially compensate a decrease of bone thickness in this place. The lamellar bone visible at the bottom of the canal (*Gruntmejer, Konietzko-Meier & Bodzioch, 2016*) is associated with marked strength of this region but accompanied by low resistance in compression (*Martin, 1991*; *Currey, 2003*; *Rhee et al., 2009*; *Achrai & Wagner, 2013*). The significant change of microstructural conditions, and thus biomechanical properties, shows that the lateral line canals might be crucial structures for the biomechanical function of the skull; especially for metoposaurids with an extremely deep system of lateral canals. The increased compactness of the bone on the bottom of the canals might be an adaptation to preserve the optimal strength of the bone with minimal thickness. However, it could have some side effect as compact bone is associated with a high strength in tension (*Martin, 1991*; *Currey, 2003*; *Rhee et al., 2009*; *Achrai & Wagner, 2013*), thus lateral canals could serve as tensile members that are subjected to axial tensile forces occurring in the skull.

The microstructure of the lacrimal, jugal and quadratojugal suggests a significantly high loading on the lateral margins of the skull. The same tendency is visible in the lateral FEA case (Fig. 7C), where an increase of stress level is suggested. It could be connected with the presence of tooth rows on the ventral side of these bones. The histological results in this case suggest more frequent occurrence of lateral biting than is concluded only from Finite Element Analysis.

With regard to the palatal side, the Finite Element loading cases suggest low stress for the pterygoid and slightly higher for the parasphenoid, which is in agreement with the microstructural results (Figs. 6 and 7). Additionally, the histological framework suggests that the parasphenoid was loaded closer to the external edges than near the central axis, where the bone is thinnest. *Marcé-Nogué et al. (2015)* pointed out that during skull raising relatively high stress affected the cultriform process. The histological results obtained herein do not appear to support the idea of any high loading on the palatal side of the skull. However, the section was done only from the very posterior part of the parasphenoid; the cultriform process itself was not sectioned. Taking into account the high microstructural variability of skull bones (*Gruntmejer, Konietzko-Meier & Bodzioch, 2016*) it cannot be ruled out that the anterior part of this bone is highly metaplastic. This, along with increased width of the process, might significantly increase the biomechanical strength of this bone. To confirm this hypothesis more sections are needed. Otherwise, the Finite Element lateral case shows a stress increase in the vomer (Fig. 7C), but this stress pattern is not supported by the histological framework, which suggests that the vomer is the weakest bone of the entire skull (Fig. 6).

The exoccipital and quadratojugal have an endochondral origin and develop via a cartilage precursor. Histologically, both bones resemble the structure of vertebrae with a highly trabecular area surrounded by a thin cortex (*Konietzko-Meier, Bodzioch & Sander, 2013*). Between the trabeculae the remains of calcified cartilage are visible suggesting low ossification of the endochondral region (*Konietzko-Meier, Bodzioch & Sander, 2013*; *Gruntmejer, Konietzko-Meier & Bodzioch, 2016*). However, in both bones

very strong Sharpey's fibres are present (*Gruntmejer, Konietzko-Meier & Bodzioch, 2016*).
Large concentrations of long, well-mineralised Sharpey's fibres in the exoccipital appear to
represent the remains of strong muscular attachments and ligaments that connected the
skull to the vertebral column that may have played a role during skull raising.

## New interpretation of mode of life

Metoposaurids were common temnospondyls confined to the Upper Triassic, with records
from several continents (*Sulej, 2007*). Despite their common occurrence and well-known
Bauplan, the mode of life of metoposaurids still remains controversial. In the past they
were either considered to have been passive bottom-dwellers in lakes and rivers, lying in
wait for prey using the passive "death-trap" model (*Ochev, 1966*; *Murry, 1989*), mid-water
feeders, comparable to temnospondyl capitosaurs (*Howie, 1970*; *Chernin & Cruickshank,
1978*; *Hunt, 1993*) or active swimmers that used limbs (*Sulej, 2007*) or tail (*Konietzko-Meier,
Bodzioch & Sander, 2013*) for propulsion.

A 3D FEA of the metoposaur skull has thus revealed that the bottom dweller and active
predator hypotheses are the best supported ones (*Fortuny, Marcè-Noguè & Konietzko-
Meier, 2017*). Metoposaurids preferred rapid bilateral biting, which according to the
present study, would confirm the ambush strategy—resting on the bottom in wait for
passing prey. The relatively low stress level found along the skull under lateral strike
indicates that lateral strike of the head was possible, even if this was not preferred and
connected with active predatory activity (*Fortuny, Marcè-Noguè & Konietzko-Meier, 2017*).
However, as mentioned above, the FEA analysis has a limitation concerning the number
of tested scenarios. Merging two different approaches (Finite Element Analysis and bone
histology) provides data from different perspectives on skull biomechanics that, when
correlated, yield a clearer image of the feeding behaviour of *Metoposaurus*. This genus
appears to have comprised aquatic animals that could adapt to various environmental
conditions and were less specialised in their mode of feeding than assumed previously
(*Ochev, 1966*; *Murry, 1989*; *Howie, 1970*; *Chernin & Cruickshank, 1978*; *Hunt, 1993*; *Sulej,
2007*). Histological results confirm the presence of direct lateral and bilateral biting, but do
not exclude other combinations, with the exception of unilateral biting, and may reinforce
the idea that lateral strike was also performed under an ambush strategy and not only
during active swimming. It cannot be ruled out that once the prey was captured by a
lateral strike, a bilateral bite was required to immobilise it, as suggested for extant giant
salamanders (*Fortuny et al., 2015*).

Bone microstructure indicates a significantly high loading on the lateral margins of the
skull, suggesting a more frequent occurrence of lateral biting than is concluded from the
Finite Element Analysis only (Fig. 7). The main biting forces are connected with long rows
of teeth along the skull margin. These rows occlude with the tooth row in the dentary,
which is supported by the presence of sharp cutting edges on the tooth margin in dentary
teeth (*Konietzko-Meier & Wawro, 2007*). Crucial is the role of the vomer tusk. As histology
reveals, the vomer is a very weak bone; on the basis of FEA, there was stress increase in the
vomer during lateral biting, but this was absent under bilateral biting. At first view, this is

contradictory. However, it may indicate that the vomer tusks only played an active role in bilateral biting, but not in lateral biting because they could easily have snapped.

The main factor that determines the mode of life may be water level. The two-season climate during the Late Triassic, with high and low water levels in local lakes and periodic rivers (*Bodzioch & Kowal-Linka, 2012*) requires changing ecological strategies to survive the unfavourable dry season. Among amphibians, the common strategy is to wait out the dry or cold season (aestivation/hibernation). However, the growth pattern preserved in long bones (revealed by histology) of *Metoposaurus* does not show distinct, seasonal Lines of Arrested Growth (LAGs) at all, but only zones and unusually thick annuli, which point to a reduced growth rate for a certain period (*Konietzko-Meier & Klein, 2013*; *Konietzko-Meier & Sander, 2013*). The numerous lines present in annulus indicate that animals reduced their activity for several short periods but did not aestivate for the entire unfavourable interval (*Konietzko-Meier & Klein, 2013*; *Konietzko-Meier & Sander, 2013*). Growth, even slow, requires a regular access to energy. Because of seasonally variable high and low water levels, feeding strategies had to be adequate to counter environmental conditions. During favourable water conditions metoposaurids may have been ambush and active predators capable of lateral strikes of the head. The dry season may have required a less active mode of life with particularly efficient bilateral biting, together with their characteristically anteropositioned orbits, optimal for ambush strategy.

Interestingly, the same feeding strategies were suggested for the small metoposaurid genus *Apachesaurus* from North America (*Fortuny, Marcè-Noguè & Konietzko-Meier, 2017*, but see *Gee, Parker & Marsh, 2017*, for a discussion of the validity of the genus). Overall it could be concluded that metoposaurids were well adapted for survival under various conditions, yet not specialised as far as feeding strategies were concerned. This ability to acquire food independently of environmental conditions could be the key character in explaining the very common occurrence of metoposaurids during the Late Triassic. However, the question remains why, in spite of their wide adaptive strategies, they disappeared, together with other temnospondyl groups, at the end of the Late Triassic. *Milner (1993)* and *Milner (1994)* documented the demise of capitosaurids, metoposaurids and latiscopids at the Norian-Rhaetian boundary as part of the end-Triassic extinction event (ETE), considered to rank amongst the 'Big Five' mass extinctions. Global changes in environmental and ecological conditions may have surpassed the adaptive capabilities for metoposaurids.

## CONCLUSIONS

1. A histological analysis of skull microstructure mostly confirms the models created by FEA, with exception of the vomer which, histologically speaking, is a low-loaded bone, but on the basis of FEA, there is stress increase in the vomer during lateral biting (absent under bilateral biting). Also, a significant change of microstructural conditions, and thus biomechanical properties, shows that the lateral line canals might be crucial structures for the biomechanical function of the skull, especially for metoposaurids

with an extremely deep system of lateral canals; this should be considered in the FEA modelling. The merging of histological studies and FEA confirm that the 'negative' scenario (in this case unilateral biting) was correctly indicated by FEA. However, the limited number of tested scenarios may erroneously interpret 'positive' behaviours and may lead to serious simplifications.

2. *Metoposaurus* was an aquatic animal that could adapt to various environmental conditions and was unspecialised in its mode of feeding. It may have used two foraging techniques in hunting; bilateral biting, as well as lateral strikes, and active hunting using lateral strikes of the head.

3. One of the potential main factors determining the mode of life may have been water level. During favourable water conditions metoposaurids may have been ambush and active predators capable of lateral strikes of the head. The dry season required a less active mode of life with particularly efficient bilateral biting; coupled with their characteristically anteropositioned orbits, this would have been optimally suited for an ambush strategy.

## ACKNOWLEDGEMENTS

We thank Hospital Mutua de Terrassa (Catalonia, Spain) for CT scanning of one of the specimens. Caio Souto Maior (UPC) is acknowledged for his help with segmentation of the *Metoposaurus* specimen. We are grateful to John WM Jagt (Natuurhistorisch Museum Maastricht, the Netherlands) for improving the English. DK-M. thanks Martin Sander for fruitful discussion and support. The journal editor (Claudia Marsicano) and reviewers (Stephan Lautenschlager and two anonymous) are acknowledged for their comments which improved an earlier typescript significantly.

### Funding

Jordi Marcé-Nogué was supported by the Deutsche Forschungsgemeinschaft (DFG, German Research Foundation, KA 1525/9-2). Josep Fortuny was supported by (1) a postdoc grant, "Beatriu de Pinos" 2014-BP-A 00048, from the Generalitat de Catalunya; (2) Spanish Ministerio de Economía, Industria y Competitividad and the European Regional Development Fund of the European Union (MINECO/FEDER EU, project CGL2014-54373-P); (3) the CERCA programme (Generalitat de Catalunya). The funders had no role in study design, data collection and analysis, decision to publish, or preparation of the manuscript.

### Grant Disclosures

The following grant information was disclosed by the authors:
Deutsche Forschungsgemeinschaft (DFG, German Research Foundation): KA 1525/9-2.
Beatriu de Pinos: 2014-BP-A 00048.

Spanish Ministerio de Economía, Industria y Competitividad and the European Regional Development Fund of the European Union (MINECO/FEDER EU): project CGL2014-54373-P.
CERCA programme (Generalitat de Catalunya).

## Competing Interests

The authors declare there are no competing interests.

## Author Contributions

- Dorota Konietzko-Meier, Kamil Gruntmejer, Jordi Marcé-Nogué and Josep Fortuny conceived and designed the experiments, performed the experiments, analyzed the data, contributed reagents/materials/analysis tools, prepared figures and/or tables, authored or reviewed drafts of the paper, approved the final draft.
- Adam Bodzioch analyzed the data, authored or reviewed drafts of the paper, approved the final draft.

## Data Availability

The skull used for FEA study and the remains after thin-sections productions of second skull together with thin-sections are housed in the collection of the Department of Biosystematics, University of Opole, Opole, Poland.

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
