# Peer review of "Merging cranial histology and 3D-computational biomechanics: a review of the feeding ecology of a Late Triassic temnospondyl amphibian"

_PeerJ, doi:10.7717/peerj.4426_

## Round 0.1 · original submission · Major Revisions

Dear Dr. Konietzko-Meier

Your Ms # 19988 entitle "Merging cranial histology and 3D-computational biomechanics: a review of the feeding ecology of a Late Triassic temnospondyl amphibian" co-authored with Gruntmejer, Marcé-Nogué, Bodzioch, & Fortuny have been reviewed by three reviewers. All concur that the your Ms needs Major Revisions before to be consider for publication in PeerJ, a decision that I completely support as Editor.

The three reviewers have commented extensively about the Ms, particularly about its unclear logical structure. The Ms needs to be reorganized in order to clearly define its aims, hypothesis to be tested and the rationale behind them. Moreover, the Introduction should be reduced and the Material & Methods expanded to include, as suggested by Reviewers #2 and #3, an accurate explanation of the FEA model used. This would certainly improve the support of the Discussion of the Ms.

A second particular issue is the missing references in the text. My concern about this point is that their inclusion will change the meaning of some of your statements, as pointed out by Reviewer #3. Accordingly, please not only include the cites suggested by the reviewers but also take particular attention to their significance in the overall structure of the Ms.

Finally, the use of the English along the Ms is sometimes problematic thus making difficult to understand the meaning of some paragraphs. Therefore, I recommend you to check the use of the language by a native English speaker before re-submission.

So, I am requesting that you revise your Ms according to the three detailed reviews enclose, taking particular attention to the points mentioned above. As the revisions required are extensive enough, another round of review may be necessary when you resubmit your revised manuscript.

Thank you for submitting your Ms to PeerJ and I look forward to receiving your revision.

Sincerely,
Claudia Marsicano

·

Basic reporting

The submitted manuscript presents an interesting approach of combining histological studies with computational biomechanical analysis using FEA. I find this approach intriguing as it can help validate results and inferences obtained from FEA studies. However, the way this is presented I this manuscript has several issues:
The introduction is detailed, but on many occasions unnecessarily long. In particular the “TESTING FEEDING” section, which spans eight pages, reads more like a review of the specific topics and would be suitable for a PhD thesis. Given that methods, such as FEA are now applied routinely in palaeontological studies, this is unnecessary and just distracts from the novelty of the methodological approach. I would recommend condensing the whole section to no more than two pages, bringing out the advantages of the presented combined methods based on how these techniques have been applied separately so far.
My biggest concern with this manuscript is the lack of detail regarding the methodology. This is particularly evident in the description of the FEA on which eight lines were spent (in comparison to the introduction of over eight pages as mentioned above). I do understand, that the author refer to a previous publication, in which the generation of the FE meshes, loading scenarios and boundary conditions are laid out in more detail. However, this study should be self-contained and the reader should not be forced to hunt down another publication to understand what has been done. Furthermore, four different loading scenarios are discussed in the results and the discussion, but only three are figured. It is very unclear what the differences in model setup between the unilateral bite and the lateral bite are, for example. Similarly, details of the model construction are not given. This is particularly important to understand, if the patters observed in the FEA models in relation to the microstructure and suture morphology are realistic or if the model has been simplified to a where these similarities are purely coincidental. As the authors state several times in the manuscript, model simplification and geometry are important factors. I would therefore expect the methods section to be very detailed.
The presentation of the whole study is OK. I have made suggestions in the attached PDF regarding spelling and grammar, but would strongly recommend to have the manuscript proof-read by a versed English or native speaker. In the results section, different tenses are often mixed throughout. Be consistent here.
Presented figures are for most parts informative and I really like how the results are presented in figure 4. I would recommend combining figures 2 (at least C and D) in some form with figure 3 so that the position and orientation of the thin-section can be recognised without going back and forth between both figures.

Overall, I think this manuscript has lots of potential and could make and important contribution as it describes a new methodological approach for biomechanical studies. That's why it is even more important to provide all necessary details to understand and possibly replicate the study design. This is not yet the case, however, and I have recommended major revisions. Fixing the raised issues shouldn't be too difficult, though.

Experimental design

The general research question is well-defined and clear. It offers a new approach to supplement biomechanical studies and to infer palaeobiological properties. Knowing the work of some of the authors I assume the experimental setup to have been performed rigorously. However, this is not visible from the described methodology (see detailed comments above). A thorough explanation of the model generation and FEA setup is very much required.

Validity of the findings

I find that the findings are mostly supported by the results. However, it is difficult to ascertain this in some aspects, as the methodology is not described in sufficient detail. This particularly relates to the setup of (apparently) four different loading scenarios, while only three are discussed and figured. This seems to be a bit cherry-picking in only showing the results that match with the histological analysis. As criticised by the authors, only a small number of different scenarios is usually tested in FEA studies. Therefore, please, figure and discuss al scenarios here as well.

Additional comments

A few further minor comments in addition to those in the attached PDF:

• Page 10, lines 102-104: I think this Is not true. A number of (biomechanical) studies have pooled various lines of information and evidence to complement functional analysis.
• Page 11, line 131: “very promising attempts” – what makes those studies promising? Did those studies fully resolve the question about feeding mode and behaviour? I would consider rephrasing this to something along the lines of “…a number of studies have used computational biomechanical analyses to address the question of…”
• Page 11, line 138: “correctness” is strong and in my opinion inappropriate word here. The correctness could only be tested if a living temnospondyl was biomechanically examined (and even then it might be difficult). Rather use “replicate the in-vivo behaviour” or similar.
• Page 20, lines 324-325: Provide institutional abbreviations directly on first occurrence. It is mentioned later in the text, but the reader shouldn’t have to search for it.
• Page 20, lines 324-329: Provide details on CT scan settings.

Reviewer 2 ·

Basic reporting

The introductions section is very long (11 manuscript pages). I have made some suggestions of background information that could be cut/reduced, for example information not directly relevant to this study (why 12 citations on sauropods and marine reptiles demonstrating they have no extant analogues?) and other sections (introducing FEA and overview of general factors impacting bone strength) what can be shortened.

On the other hand, some very relevant findings from previous studies involving sensitivity analyses in FE studies of other fossil taxa and/or validation studies on FE models of crocodilian skulls (similar in both shape and ecological niche to temnospondyls) are not mentioned at all. Similarly, although correlating FE results with histology is novel, doing so with cranial sutures is not (Line 137) – see extensive work by Rayfield, Moazen Reed and others in fossil and living reptiles and fossil tetrapods.

It is unclear to me (see comments Line 304) why an aquatic predator cannot be an ambush predator using a lateral strike, as this occurs in some extant aquatic predators.

I have made suggestions (see “Validity”) on one figure that can be modified slightly and one figure that could be added that would clarify results.

The English language needs to be improved throughout the manuscript to ensure your international audience can understand your text. I suggest having an English speaking colleague review your manuscript. I have made some corrections and highlighted difficult text (any highlighted text without additional comments are areas where the language is unclear) throughout the annotated PDF.

Experimental design

One of the issues with this manuscript is a lack of clear questions that will be addressed or hypotheses that will be tested (stating how results from FEA and bone microstructure would support or reject those hypotheses). You must add such a section at the end of your introduction.

I realize that the detailed methods for the model have been published elsewhere but basic information needs to be repeated here – see annotated PDF line 371. This is especially true if – as you suggest – you have made changes to your model since its 2017 publication. Any such changes need to be detailed in full to allow the study to be replicated.

Your text (Lines 529 – 531) suggests the lateral strike loading regime is not the same as a unilateral bite. If not, please explain (early on when you discuss results from your 2017 paper) what a lateral strike load regime actually involves and how it differs from unilateral biting.

Validity of the findings

One thing that would help tremendously in understanding the Discussion section “Bone histology vs computational biomechanics – how does bone structure correlate with Finite Element Analysis?” would be if you could overlay bone boundaries onto the screenshots of FE models showing von Mises stress (Figure 8). Readers could compare these with the labelled maps of the skull shown in Figure 2 to understand within which bones different stresses are occurring under different loading conditions.

Also – why not have contour plots of compressive vs. tensile stress to see whether your predictions on stress type based on sutural morphology corresponds with FEA predictions? Von Mises stress plots do not add information beyond stress magnitudes, whereas suture shape should correlate with the type of stress (tension, compression) being experience.

Be careful with the language you use in the final section of the discussion (new interpretation of mode of life) – although you have evidence to support your suggestions, they are still speculation. Some of the wording you use makes these ideas sound conclusive.

Additional comments

The study being proposed here – in terms of using bone microstructure as a reality check on the predictions of finite element models – is genuinely novel, interesting and worthwhile, and I would like to see a revised version of this paper in print. There are some findings that I see as particularly interesting that should be highlighted more - particularly the impact of the deep and extensive lateral canals on skull mechanics in Metoposaurus and bone microstructure/FE results indicating that some bones are resistant to stresses incurred during head raising (rather than biting), which is not a commonly tested load regime.

However, there are numerous issues (see above) with the manuscript that need to be overcome prior to publication.

Annotated reviews are not available for download in order to protect the identity of reviewers who chose to remain anonymous.

Reviewer 3 ·

Basic reporting

The paper contains some interesting data on bone histology and how histological features change across the skull of the Late Triassic temnospondyl Metoposaurus. However, I found the main aim of the paper was difficult to determine. What is the hypothesis/hypotheses being tested here? Is it that cranial histology can be used to predict skull stress? Is it that increased bone compactness and increased bone thickness are related to increased skull stress? Is it that cranial sutures can be used to predict cranial strain? Is it to test the null hypothesis that Metoposaurus was an active hunter, or passive bottom dweller? The hypothesis / hypotheses being testing need to be clear from the start of the paper.

The introduction is very long. It would benefit from sharpening and tightening the text. This will be made easier if the hypotheses are clearer. Also there are sections of text that are basic background information that don’t need to be included in a manuscript – a good example is lines 192 – 227 on bone structure. It would be appropriate to include such information in a PhD thesis for example, but it’s just not necessary to include such detail on basic bone biology.

Unfortunately, although the introduction is long, the authors fail to cite some key papers in the field. On line 136 it is stated that “Thus in the present study is for the first time correlation of histological and cranial suture morphology results with computational biomechanics (FEA) is done”. This is simply not true. There are lots of papers that compare histology and suture morphology to the results of FE-analysis – work by Moazen, Porro, Jasinoski, Rayfield on extinct and extant taxa – there’s a long history of papers in the this area but different research groups and they are not cited at all here. Some examples are: Rayfield 2004, 2005; Jasinoski et al. 2010a,b, Jasinoski & Reddy 2010; Moazen et al. 2009, 2013; Porro et al 2013, 2014; Curtis et al. 2013; Jones et al. 2011) . These papers need to be acknowledged and the statement rectified.

The paper would benefit from a check on the English language by a native speaker – see quoted text above as an example. There are many sentences that require rephrasing. This is concerning because I’m not aware if PeerJ editorial team will make such changes in house.

Experimental design

The authors outline why they are focusing on Metoposaurus, but if the aim is to demonstrate how cranial histology can be used to predict skull stress and strain, then why not complete the study on an extant animal where cranial strain can be measured experimentally ex vivo or in vivo, and then compare these measurements to bone histology? Whether the authors need to address the issue of why extant datasets were not used depends on what this paper is trying to achieve. If it is to document histological features across the skull and compare these to FEA results, then fine. But if it is to introduce bone compactness and thickness as a proxy for predicting skull stress and strain, then the authors need to address why they have not considered used extant animals as a case study.

Bone thickness and bone compactness are measured. However, on what basis do these two variables, either independently or combined, act as useful indicators of bone loading? Is this based on common sense logic – bones should be thicker and more compact where stresses are higher in magnitude? Or is this based on quantified data from living animals and associated published work? Please state why you used these metrics, and why they are appropriate stress indicators?

FE models. The FE models used in this paper have be published previously. However this manuscript needs to include much more information on model construction. What were the boundary conditions of the model – loadings and constraints? What material properties were used? How many elements etc. Yes, a reader can refer to the previous paper, but this current manuscript needs to contain sufficient information to be a standalone publication.

Validity of the findings

Sutures. If all the sutures in the skull are recognised as interdigitated, then why go to the effort here to talk about different types of sutures? And how does suture morphology link to skull strain? Granted, the von Mises plots will show the location of peak strain in this homogenous and linear elastic model, but von Mises doesn’t tell you whether the stress and strain is tensile or compressive. Do interdigitated sutures in the skull of Metoposaurus follow the predictions for the types of strain experienced across interdigitated sutures? IE are all the strains compressive across the interdigitate sutures? If not, then where does this leave the role of sutures as predictors of cranial strain? Do we believe the signal from the sutures or the signal from the models?

I felt that similar ambiguity was present when discussing the results of the comparison between bone compactness and thickness and skull stress. Figure 8 nicely illustrates the spatial relationship between FE-derived stress and microanatomy-derived stress, but I was looking for a conclusion to this work – which of these two metrics is the better proxy for biomechanical behaviour? But again, this relates to my comments at the start of the review – what is the manuscript trying to achieve? Is it trying to compare these two methods for determining skull stress? If so, how do you reconcile when the two types of data give conflicting results. Or is it attempting to work out the feeding behaviour of Metoposaurus? The discussion focuses more on the latter issue.

Additional comments

My overarching comment to the authors is that they need to specify what this paper is trying to achieve. Once this has been determined, everything else will flow much better – superfluous information can be cut from the introduction and the results and the discussion can be targeted better. And the authors must acknowledge the large body of work that has already attempted to link suture morphology to FE-derived stress and strain patterns.

---

## Round 0.2 · Minor Revisions

Dear Dr. Konietzko-Meier

Your Ms # 19988 entitle "Merging cranial histology and 3D-computational biomechanics: a review of the feeding ecology of a Late Triassic temnospondyl amphibian" co-authored with Gruntmejer, Marcé-Nogué, Bodzioch, & Fortuny have been reviewed by two reviewers and myself.

I certainly concur with the reviewers that your Ms has substaintially improved since its first version. Nevertheless, some minor changes are needed before final acceptance.

Please, pay particular attention to the reviewers comments about not only some rephrasing (Reviewer #1) but also the structure of the "Introduction" (Reviewer #2) in order to make the text more clear and fluent. In this context, they both also included several minor changes in the annotated versions of you Ms attached.

Also, the reviewers have pointed out some missing data in the FE method section (Reviewer #2) and in Figure 2 (Reviewer #1) that need to be addressed.

Finally, I agree with both reviewers that the language is definitely better than its first version but still needs some improvement. Therefore, I strongly suggest that you to check it by a native English speaker before submission.

I am requesting that you revise your Ms according to the reviews enclose and paying particular attention to the points mentioned above.

Thank you for submitting your Ms to PeerJ and I look forward to your final revision.

Sincerely,
Claudia Marsicano

·

Basic reporting

I find the revised version very much improved in comparison to the original submission. Suggestions and comments from myself and the other reviewers have been incorporated into the revised manuscript making the whole study more accessible to the reader.

Experimental design

See above

Validity of the findings

See above

Additional comments

I have a few minor comments and remarks listed below: In addition I have made some grammatical suggestions in the attached document in the form of track changes. The language of the manuscript has considerably been improved, but if possible another check by a native English speaker would be recommended.



Abstract

• “To test this scenario, the early tetrapod group of Temnospondyli is an excellent case-study” – Some brief explanation is necessary here to support that statement.
• “However, microscopic data suggest that…” – What is meant by ‘microscopic data’ here and on other occasions in the text?
• Results in the abstract are very much focussed on the reconstruction of the palaeobiology of Metoposaurus, but there is barely any word about the methodology and how reliably histology can replicate the FEA results and vice versa. The latter is, in my opinion, the more interesting subject of the manuscript and should be mentioned in more detail in the abstract.
Introduction

• “For fossils taxa without modern analogues these variables moreover have to be assume based on the taxa with a similar Bauplan, but sometimes far phylogenetically (Anderson et al., 2012) as it is i.e. for Temnospondyli where commonly the Crocodylus is used as a proxy (see Fortuny et al., 2016 for a discussion).” – It is not clear what is meant by ‘far phylogenetically’. Consider simplifying and rep[hrasing that sentence.
• “It drive to make assumptions regarding bone properties and moreover calculations are simplified (see Bright, 2014; Fortuny et al., 2015 and references therein for discussion ).” - Not clear what ‘it drive’ means. Consider rephrasing.
Discussion
• “A slight increasing of stress level in the FEA model of skull raising (Fig. 7D reinforces the support to this hypothesis” – Rephrase sentence, meaning not clear.

Figures
• Figure 2 caption is missing indices (A, B, C) for the different scenarios. In addition more detail is necessary explaining the different parts of the figure (e.g. load forces and scenarios)

Reviewer 2 ·

Basic reporting

The authors have made substantial improvements to the manuscript since the first version and they are to be particularly commended on greatly reducing the overall length of the Introduction (especially the FEA section) and improving it’s focus, adding clear study goals at the end of the introduction, the much more detailed methods section – particularly the careful description of the different loading cases and Figure 2, which is excellent and very clear, which muscles were incorporated into the model, and the nature of boundary conditions and material properties – as well as the excellent new Figure 7.

There are still minor revisions needed prior to publication.

The introduction could have a more orderly flow – at the moment, the authors 1) introduce FEA and simplifications that must be made in fossil taxa such as Temnospondyls, 2) introduce bone histology and how it may be related to mechanical loads, 3) introduce the most common bone histology found in temnospondyls, 4) formally introduce temnospondyls, 5) review previous FE studies on temnospondyls, 6) discuss more bone histology in temnospondyls, and finally 7) discuss study goals. It may be more straightforward to rearrange the sections, for example: 1) introduce temnospondyls and uncertainties of their feeding modes, 2) introduce FEA, review previous FE studies on temnospondyls, and discuss assumptions that must be made for fossil taxa, 3) introduce the idea that bone histology and feeding loads are related and discuss specific histology (and what this might mean for feeding loads) in temnspondyls, and 4) finally bring it all together in the study goals – they will compare data from FEA and bone histology to test the accuracy of predictions from FEA and try to uncover feeding mode.

The language in the introduction, results and discussion still needs improvement to ensure your international audience can understand your manuscript. I have made corrections until the end of the Introduction section of the text.

Experimental design

On several occasions (in the Abstract starting line 34, Introduction starting line 136) the authors refer to the taxon as an “excellent case study”. This is somewhat of an exaggeration – as pointed out by Reviewer 3 during the first round of reviews, it would be ideal to test out this idea on extant animals first. Metoposaurus is interesting because it is very common (and thus there are specimens to be sectioned) and its feeding mode is debated – I think the language here just needs to be toned down a bit.

The FE methods sections is greatly improved. One important detail that is still missing are the magnitudes of the applied muscle forces. These could be stated in line with the text, for example at Line 225 before the end of the sentence you could add: “…this resulted in applied muscle forces of XX Newtons for the AMI, XX Newtons for the AME and XX Newtons for the AMP.” This is important data in order to replicate the study. Also, what were these magnitudes based on – reconstructed cross-sectional area?

One last question – wouldn’t the metoposaur need to also bite down (unilaterally) on its prey during a lateral strike, in addition to pulling the prey sideways? Maybe add a line to justify modelling a lateral pull without the animal unilaterally biting down.

Validity of the findings

No comments.

Annotated reviews are not available for download in order to protect the identity of reviewers who chose to remain anonymous.

---

## Round 0.3 · accepted · Accept

Dear Dr. Konietzko-Meier

I am pleased to inform you that your manuscript # 19988 entitle "Merging cranial histology and 3D-computational biomechanics: a review of the feeding ecology of a Late Triassic temnospondyl amphibian" co-authored with Gruntmejer, Marcé-Nogué, Bodzioch, & Fortuny is now accepted for publication in PeerJ.

Thank you again for considering PeerJ and we look forward to your future contributions to the Journal.

sincerely,

Claudia Marsicano